# Material Extrusion-Based Additive Manufacturing of Poly(Lactic Acid) Antibacterial Filaments—A Case Study of Antimicrobial Properties

**DOI:** 10.3390/polym13244337

**Published:** 2021-12-11

**Authors:** Piotr Gruber, Viktoria Hoppe, Emilia Grochowska, Justyna Paleczny, Adam Junka, Irina Smolina, Tomasz Kurzynowski

**Affiliations:** 1Center for Advanced Manufacturing Technologies (CAMT-FPC), Faculty of Mechanical Engineering, Wroclaw University of Science and Technology, 50-371 Wroclaw, Poland; piotr.gruber@pwr.edu.pl (P.G.); emilia.grochowska@pwr.edu.pl (E.G.); iryna.smolina@pwr.edu.pl (I.S.); tomasz.kurzynowski@pwr.edu.pl (T.K.); 2Faculty of Mechanical Engineering, Wroclaw University of Science and Technology, 50-371 Wroclaw, Poland; 3Department of Pharmaceutical Microbiology and Parasitology, Wroclaw Medical University, 50-556 Wroclaw, Poland; justyna.paleczny@student.umw.edu.pl (J.P.); adam.junka@umw.edu.pl (A.J.)

**Keywords:** biomaterials, filaments, antimicrobial, fused filament fabrication (FFF), fused deposition modelling (FDM), Poly(lactic acid) (PLA)

## Abstract

In the era of the coronavirus pandemic, one of the most demanding areas was the supply of healthcare systems in essential Personal Protection Equipment (PPE), including face-shields and hands-free door openers. This need, impossible to fill by traditional manufacturing methods, was met by implementing of such emerging technologies as additive manufacturing (AM/3D printing). In this article, Poly(lactic acid) (PLA) filaments for Fused filament fabrication (FFF) technology in the context of the antibacterial properties of finished products were analyzed. The methodology included 2D radiography and scanning electron microscopy (SEM) analysis to determine the presence of antimicrobial additives in the material and their impact on such hospital pathogens as *Staphylococcus aureus*, *Pseudomonas aeruginosa,* and *Clostridium difficile*. The results show that not all tested materials displayed the expected antimicrobial properties after processing in FFF technology. The results showed that in the case of specific species of bacteria, the FFF samples, produced using the declared antibacterial materials, may even stimulate the microbial growth. The novelty of the results relies on methodological approach exceeding scope of ISO 22196 standard and is based on tests with three different species of bacteria in two types of media simulating common body fluids that can be found on frequently touched, nosocomial surfaces. The data presented in this article is of pivotal meaning taking under consideration the increasing interest in application of such products in the clinical setting.

## 1. Introduction

The Food and Drug Administration (FDA) reporting activities indicated that the COVID-19 epidemic would affect the supply chain of medical products and supply disruptions or shortages of critical medical products [1,2,3]. In the context of the COVID-19 pandemic, the choice of fillers in polymers is, to a significant extent, dictated by copper’s strong antiviral and antibacterial effect [4]. In addition, also due to the increased production of personal protective equipment (PPE) and other critical medical materials and devices, the demand for additive technologies has increased [5]. All surfaces should be considered at risk of contamination by microorganisms transmitted from the patients, visitors, and medical personnel in the clinical setting. Microbes (viruses, bacteria, and fungi) can also be dispersed by aerosols or body fluids on medical surfaces. Therefore, biocidal surfaces are essential in preventing so-called nosocomial (hospital) infections and epidemic outbreaks [6]. Palza [7] suggests that the addition of copper nanoparticles to polymers, thanks to their antimicrobial properties, is a promising application for the development of medical devices protected from microbial contamination. In the polymer–Cu composites, there is a similar tendency, dependent on the number of particles in the polymer, directly related to the release of the metal ions responsible for the antimicrobial effect [8]. 

Poly(lactic acid) (PLA) is a low-crystalline, biodegradable polymer widely used numerous of industry fields [9]. PLA due to its innate characteristics is also commonly used in 3D printing technology, especially considering its ease of production and source material availability [10]. As an additive manufacturing method, Fused Filament Fabrication (FFF), also known as Fused Deposition Modelling (FDM), enables rapid prototyping and on-demand production [11]. Thanks to the properties mentioned above, PLA is used to produce medical equipment, such as PPE, needed to keep healthcare personnel, patients, and public service employees safe during the COVID-19 pandemic [10]. The combination of the production capabilities of FFF devices, along with the availability of materials and open-source digital models, allowed for a quick response during a pandemic crisis and immediate production of the necessary products [12]. 

Applications such as medical and biomedical as well as food packaging would benefit from antibacterial properties acquisition [9,13,14]. There are two strategies for the design of antimicrobial materials. The basic approach is to change the chemical properties by adding active agents to the polymer matrix, which are usually added on mesoporous silica nanoparticles carriers [15]. Typical antimicrobial additives or fillers reported in literature are Ag, Cu, Zn, ZnO, TiO_2_, MgO, or SiO_2_ [16,17] as well as blends consisting of ceramics—Al_2_O_3_ or SiO_2_ in combination with active nanoparticles [18,19]. Another way is to modify the surface by adding geometric patterns on it that reduce bacterial colonization of the surface. Due to the limitations of the developed surface topography, the contact area of bacteria is narrower compared to the condition of the smooth surface. And it is possible to use the synergistic effect of both methods, which may lead to even better antimicrobial properties [20]. Research on a commercial product for FFF technology applications showed that PLA with a 1% addition of copper nanoparticles was up to 99.99% effective against *S. aureus* and *E. coli* after a 24-h incubation period [21]. PLA, as an additively manufactured material, is one of the most widely used polymers in medicine applications [22].

Therefore, the subject of this article is the verification of commercially available filaments, whose manufacturers claim antimicrobial properties of their product. The lack of antibacterial properties may lead to the inappropriate use of these materials or mislead a potential customer. The verification was done on two levels: (1) analysis of the additives’ presence in filaments using 2D digital radiography and SEM and (2) biological tests to analyze the antimicrobial properties of filaments. The biological tests were carried out using three different bacterial species (*S. aureus*, *P. aeruginosa*, and *C. difficile*), considered to be the prevalent etiological factors of hospital infections. 

The novelty of the results consists of considering the potential antibacterial properties declared by the manufacturer, represented by details produced using FFF technology, and verifying them to our own research, using other than provided in the standard bacterial species, concerning potential medical applications.

## 2. Materials and Methods

### 2.1. Materials

Nine antibacterial filaments with a diameter of 1.75 mm from commercial producers were evaluated (Table 1). The base material of filament is PLA. The antibacterial additives are copper (Cu) or silver (Ag) in different forms, such as nano- or micro-particles. Each manufacturer used a different type of additive (Table 1). According to manufacturers’ claims, all materials displayed strong antibacterial properties.

### 2.2. Sample Manufacturing

The samples were manufactured with Fused Filament Fabrication technology. Prusa i3 MK3S+ (Prusa Research a.s, Prague, Czech Republic) device with a 0.4-mm nozzle diameter was used. The manufacturing process was prepared in PrusaSlicer 2.3.0 (Prusa Research a.s, Prague, Czech Republic) consisting of 30 cylindrical biosamples (ø10 mm × 2 mm) and a single specimen dedicated for 2D digital radiography analysis (2 mm × 10 mm × 2 mm). Each material was processed separately using temperature experimentally selected from the range given by the materials’ manufacturer. The process parameters, along with temperature settings, are presented in Table 2. 

### 2.3. 2D Digital Radiography

The test samples, a piece of filament (ø1.75 mm × 35 mm) and 3D printed (FFF) specimen (2 mm × 10 mm × 2 mm) were analyzed using the digital radiography technique, which allows for a non-destructive qualitative assessment of the internal features of the tested objects. Furthermore, it is a tool for quick and therefore cost and time-efficient verification of samples [23,24]. For this purpose, a 300-kV microfocus X-ray tube and a high-contrast digital flat panel detector GE DXR250 (GE Sensing & Inspection Technologies GmbH, Wunstorf, Germany) was used. X-rays were taken using constant and repeatable parameters for all analyzed filaments: voltage of 70 kV, current of 90 μA, the integration time of 1000 ms, and a magnification of 130×. Additionally, the detector was calibrated before each measurement to avoid image distortion caused by constant pattern noise caused by differences in detector components and electronics.

### 2.4. Scanning Electron Microscopy (SEM) Analysis

In order to determine the presence of additives in the filament, the cross-sections of all filaments were performed. The samples (ø1.75 mm × 35 mm) obtained in this way were sputtered with a gold layer to obtain a conductive layer enabling better imaging. Then, they were examined using a scanning electron microscopy (SEM) Sigma VP 500 microscope (Zeiss, Oberkochen, Germany) in HDBSD imaging mode. Point analysis was performed using the Octane Elect EDS System detector (Ametek, Mahwah, NJ, USA), with the acceleration voltage 20 keV and working distance WD = 8.5 mm. 

### 2.5. Antimicrobial Properties

The assessment of the ability to colonize the surface by microorganism/potential antimicrobial activity was carried out in the standard Tryptoc Soya Broth (Biocorp, Warsaw, Poland) medium (System I) or in artificial saliva (AS) (System II) with the following composition: an aqueous solution containing 2.5 g of mucin, 0.25 g of sodium chloride in 1 L, 0.2 g of calcium chloride; 2 g of yeast extract, 5 g of peptone, and 1.35 mL of 40% urea [25]. After introducing the ingredients and mixing, the artificial saliva was adjusted to a pH of 6. The microbiological tests against the Gram-positive *S. aureus* ATCC 6538, non-toxicogenic strain *C. difficile* BAA 1801 and Gram-negative *P. aeruginosa* ATCC 15442 bacteria, were performed using:(1)a standard method to assess the level of biofilm formation using the non-specific ability of crystal violet to bind to bacterial biomass [26];(2)a method using the reduction of colorless tetrazolium chloride to red formazan crystals in the presence of the living, metabolically active microorganisms [27];(3)quantitative cultures in an anaerobic atmosphere [28].

Research using techniques (1) and (2) was carried out for *S. aureus* and *P. aeruginosa*; technique (3) was used towards *C. difficile*. As a control surface, polypropylene (PP) specimens with a diameter of 10 mm and a height of 1 mm were used for cell culture tests; (1) and (2) were performed in 6 replications; tests (2) were performed in triplicate. The reduction of bacterial cells grown on the test surfaces compared to the cells grown on the control surfaces was calculated using the following Equation (1): (1)$$100\%-\left(\frac{absorbance\;defined\;for\;analysed\;samples}{absorbance\;defined\;for\;control\;samples}\right)\times 100$$

Or, in case of quantitative culturing, according the following Equation (2):(2)$$100\%-\left(\frac{number\;of\;colony-forming\;units\;defined\;for\;analysed\;samples}{number\;of\;colony-forming\;units\;defined\;for\;control\;samples}\right)\times 100$$

## 3. Results and Discussion

### 3.1. 2D Digital Radiography

Images obtained with the radiographic method for filaments and additively produced samples are compiled in Table 3. Images obtained in the radiography technique are in grayscale, corresponding to the density of the scanned material. Therefore, the background and voids are seen as relatively bright areas, while the higher density areas are seen as darker in the grayscale. It can be observed that the filaments do not show visible internal voids. The inclusions of materials of higher density than polymer matrix are recorded instead. These fillers are expected to improve the antimicrobial properties of tested materials. In addition, most of the materials show a characteristic content of additives with an even distribution over the verified height. The exceptions are A and D filaments, which, as revealed by image analysis, did not show the content of an additive with a higher density than the base material. This may suggest using a dispersion filling that does not allow registering the footprint of antibacterial additives at a given resolution or suggesting an uneven additive distribution over the entire mass of the filament spool. The samples produced in the FFF technology from the tested filaments show similar results. In addition to additives and their unequal distribution, the produced samples show defects and discontinuities due to the disconnection of the layers during the FFF process. Despite the use of the same process parameters, the dedicated printhead temperatures and bed temperatures of the sample differ from each other. This behavior can be explained by differences in melt flow rate for a given material in a selected processing temperature. The worst filling effect was obtained for A, B, and I.

### 3.2. SEM–EDS Analysis

The backscattered electrons (BSE) imaging mode enabled contrast imaging of the difference in material density. Results are presented in Figure 1. No additions were observed in the SEM images on the cross-section of samples of A, B, and D. Areas with a different density than the matrix were observed in the highest manner in samples E and G. Material I was characterized by high porosity (black areas) and the presence of metallic additives. On the other hand, a small concentration of the metallic additives was observed for materials C, F, and H.

It should be emphasized that in the case of SEM–BSE, the bright particles corresponded to the material of a higher density (ρ_PLA_ = 1.25 g/cm^3^, ρ_Ag_ = 10.49 g/cm^3^, and ρ_Cu_ = 8.96 g/cm^3^). Only 5 out of 11 tested filaments had a specific metallic additive—copper or silver. In the remaining six cases, the manufacturers did not specify what material the antibacterial additives are. Therefore, it was decided to carry out a point analysis of the places indicated by the arrows. Due to the availability of additive (J) used in E-material used by the manufacturer, it was decided to present the SEM–EDS analysis of this additive. 

Literature indicates that such nano additives as Ag, Cu, ZnO, TiO_2_, MgO, or SiO_2_ are successfully used as antimicrobial components against popular species of bacteria, in particular *S. aureus* and *E. coli* [16]. Moreover, it is popular to use mixtures consisting of ceramics—Al_2_O_3_ or SiO_2_ in combination e.g., with silver nanoparticles [18,19]. According to the manufacturer’s declaration, the SEM–EDS analyses confirmed that the G-material contains Ag, and the material C and E, respectively, contains Cu. The detailed results of the point analysis of micro-areas with different phase contrast presented in Figure 1 are presented below (Table 4).

All analyzed materials, due to the declared presence of nano additives or other metallic additives, which are usually dispersed in the ceramic support, contain Al, Si, and O, which leads to the reflection that these elements form silica or alumina compounds. Moreover, in the case of materials B, D, and F, the presence of Mg was found, which can form the MgO compound of antibacterial properties according to the relevant literature data [29]. The attention should be paid to the Ti present in the A, B, D, F, and I materials. Apart from the antibacterial properties of the TiO_2_ compound [30], its presence may also be dictated by the role of the dye in these filaments. Among the analyzed materials, the filament from E and H manufacturers is characterized by the highest purity, i.e., the absence of any other chemical elements than Cu.

### 3.3. Microbiological Properties of Tested Materials

The biological tests had shown that when standard microbial TSB medium (which conditioned the surface of analyzed materials) was applied, in specific cases, the higher colonization of bacteria was observed on analyzed than on control samples, devoid of antimicrobial additive (Figure 2). Especially in the case when *P. aeruginosa* was scrutinized, there was no significant decrease in bacterial colonization on the surfaces, particularly when the XYZ Printing material was applied. In turn, the bacterial survival rate ranged between 90 and 95% indicated that such materials as G, B, D, A, or E do not show significant antimicrobial activity within the experimental setting used. The most significant reduction of bacterial cells was observed for *S. aureus* regardless type of surface applied. The level of bacterial survival ranged from about 40% for G material to 63% for F material. In the case of *C. difficile*, the survival rate of bacterial cells for most materials was ca. 60 to 80%. When the second system (based on artificial saliva) was used (Figure 3), a similar trend in the behavior of bacterial cells was observed. 

In the case of *P. aeruginosa*, incubated in the medium-I, the growth of bacterial cells was higher than in the control setting. The most extraordinary stimulation was observed for G, E, and I material, where the number of bacteria compared to the control was over 100%. An insignificant decrease of cell number (5–20%) for F, A, D or C, and H was observed. A higher decrease of *S. aureus* cell number (in comparison with *P. aeruginosa*) was detected (30–60%) for the aforementioned materials. In turn, *C. difficile* displayed differentiated susceptibility to applied materials (5–60% reduction rate, dependent on surface, Figure 3). The highest reduction for species above was recorded towards A-material (60%). 

Thus, the obtained results (Figure 2 and Figure 3) show that not all tested materials after processing in FFF technology display the expected antimicrobial properties declared by the materials’ manufacturers.

Confronting the obtained results with the data provided by the manufacturers, it can be observed that the producers mainly rely on the ISO 22196 method for testing the antimicrobial activity of plastics. According to the mentioned standard, research is conducted against *E. coli* and *S. aureus* usually in a long, 24-h contact time (Table 5). 

These tests, however, do not reflect the real conditions faced by PLA materials, which are exposed to the recurring heavy burden of plethora of various nosocomial pathogens to which *P. aeruginosa* and *C. difficile* also belong. Moreover, there are important etiological factors of a wide spectrum of disease entities, including infections of respiratory, alimentary, urinary tract and skin, soft tissue, bones, and gut [31,32,33,34]. It is noteworthy that the 24-h contact time (period of exposure of microbes to additives included to surface of material) seems to be rather irrelevant concerning the potential application (face masks, door openers) of PLA materials in a hospital setting due to the frequency of real use of the PLA-finished products is decisively higher. Rather than searching for implementation in shared space equipment and public areas, emphasis should be put on parts used during the provision of hospital care or by a single patient. The approach represented in this manuscript, which is not only based on the guidance given by the standard, has also been implemented by Thavomyutikarn et al. [35]. These studies are based only on Gram-positive bacteria—*S. aureus*, *S. epidermidis*, and *B. subtilis*. 

Interestingly, Maróti et al. [36] dealt with the problem of PLA processing and the analysis of the antibacterial effect on these materials, using the following bacterial species: *Micrococcus luteus—Sarcina, Bacillus subtilis, Staphylococcus aureus, Escherichia coli,* and *Pseudomonas aeruginosa*. This approach confirms that it is also worth analyzing other bacterial species that constitute the skin bacterial microbiota and those that cause severe respiratory and urinary tract infections. The approach is consistent with the standard for the analysis of PLA material and was represented by other research groups, Mania et al. [37] and León-Cabezas et al. [38], which, following the guidelines, analyzed the antimicrobial response using bacterial *E. coli* and *S. aureus* species. FFF technology can also influence the filler distribution in terms of antimicrobial properties, which may affect its final properties. Unfortunately, materials’ manufacturers do not always provide complete information about the conducted microbial test. Thus, there is no certainty if tests were performed on raw feedstock material (filament) or manufactured samples. 

The other common issue is not providing the chemical composition of materials by manufacturers. It is understandable from the “know-how” perspective; however, it also significantly decreases the possibility of deducing how the given materials may work or be applied in the specific, actual clinical situation. We are convinced that in the pandemic period, the basic chemical composition should be revealed for the sake of customers. Another issue is a lack of a common standard for the antimicrobial tests of polymer samples. For example, certain manufacturers in materials datasheets present results of such testing after 8 h. The others show datasets and changes for measurements for 6, 8, and 24 h. The lack of clearly defined rules makes assessing the antibacterial effectiveness between individual filament samples impossible or significantly challenging. Particular attention should be paid to the time intervals between subsequent disinfection due to the use of such materials in the hospital practice. For example, as dedicated material for door openers, during the coronavirus pandemic, hospitals point out the problem of the disinfection of basically all places that can be touched by hospital staff, patients, or visitors. Therefore, the question that should be addressed and answered is how effectively the antimicrobial additives act within shorter times; the results of our case study indicate that their activity within such contact time is rather scanty. We are aware that extrapolation of data shown in this study in actual clinical settings should be taken with precaution. Further experiments (shorter contact times, higher number of strains scrutinized) should be performed to verify our discovery’s importance. Nevertheless, we are convinced that our article may be considered an essential step in applying PLA products of high antibacterial activity. 

## 4. Conclusions

The approach includes tests that exceeded the test scope dictated by the standards and significantly expanded state of the art regarding antimicrobial commercial polymer materials used in FFF printing. Due to the COVID-19 pandemic, the demand on antimicrobial surfaces is needed. Not all commercially available surfaces declared as “antibacterial” in our study meet the actual clinical requirements concerning their antimicrobial activity. Higher availability of data on materials’ composition and testing details should be provided during a pandemic crisis. 3D printing technologies enable the use of antibacterial materials and a quick response to immediate prototyping of elements for hospitals and other public utilities in the crisis caused by COVID-19. Activities should focus on the development of antibacterial materials used in FFFs, which can be successfully used in low-volume and specialized production. Commercial research performed by laboratories focuses on ad hoc material analysis. However, an essential factor is the process of additive processing itself, which is often overlooked in such analyses. 

The development of commercially available antimicrobial polymers for additive manufacturing has enabled the prototyping of a wide variety of critical medical devices by many printers worldwide. Although the materials used were to be antibacterial, in most cases, these polymers do not meet the expectations and criteria that should be set for hospital use. Therefore, for commercial use in the specific hospital environments and other peri-medical applications, these materials should undergo a series of other tests on dedicated demonstrators.

## Figures and Tables

**Figure 1 polymers-13-04337-f001:**
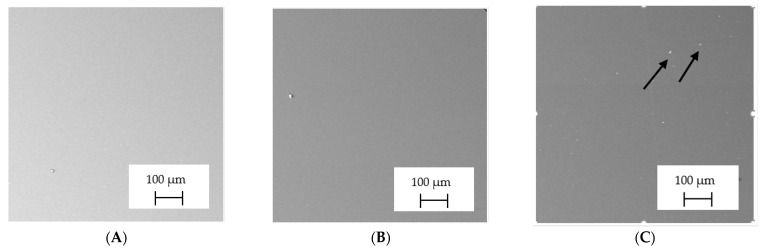
Results of SEM–BSE cross-area observations. Black arrows indicate exemplary particles of additives. (**A**–**I**): specific type of material analyzed. (**J**)—additive used in material E.

**Figure 2 polymers-13-04337-f002:**
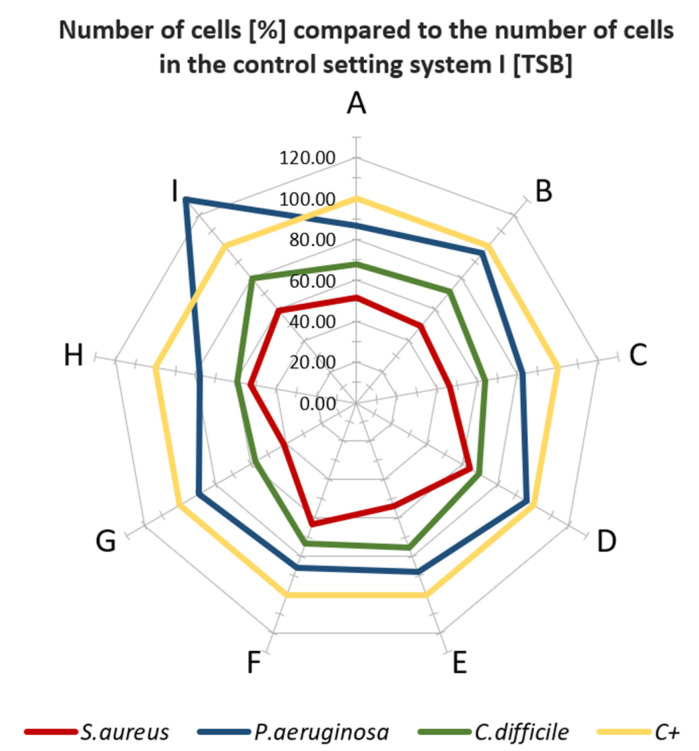
Plot of number of viable cells (%) on the surface of the test samples compared to the number of cells grown on the control surfaces in test setup I. Letters A–I: specific type of material applied.

**Figure 3 polymers-13-04337-f003:**
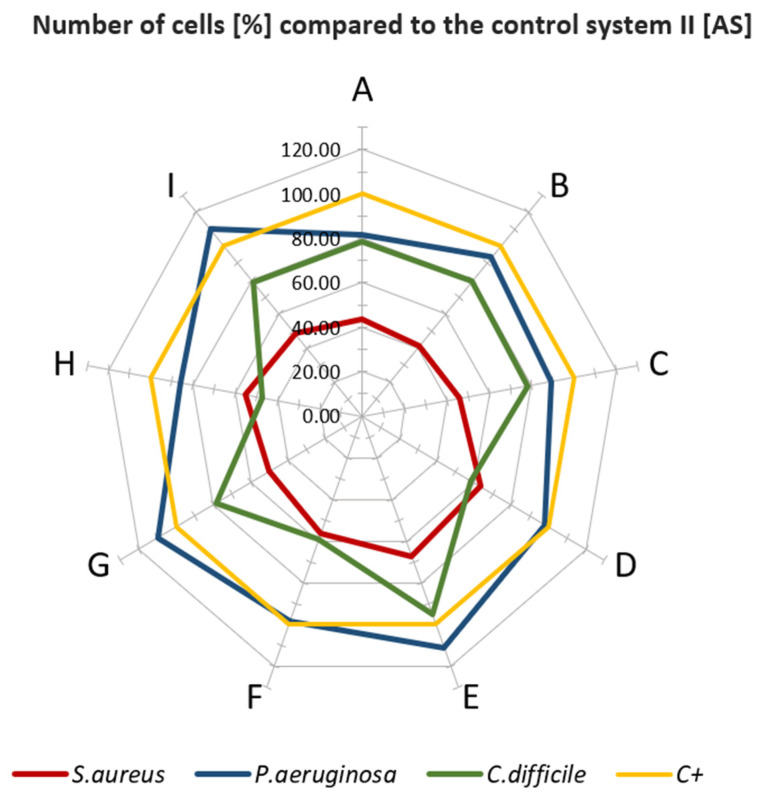
Plot of number of cells (%) compared to the control system II [AS]. Letters A–I: specific type of material applied.

**Table 1 polymers-13-04337-t001:** The list of filaments with additions.

Name in the Manuscript	Trade Name	Manufacturer	Chemical Composition	LOT; S/N; Batch No.
A	AbFil PLA 850	3D Fils (Elche, Spain)	PLA with silver additives	20042908DIJ
B	Mega 3D Antibacterial PLA	FiberForce (Treviso, Italy)	Based on PLA, manufacturer doesn’t specify additives	FX-100-30
C	NanoCICLA	Cicla3D (Bío, Argentina)	PLA with copper nanoparticles	0000000503
D	PLA Antibacterial	Philament/Filaticum (Miskolc, Hungary)	PLA with metal additives	N/S *
E	PLActive AN^1^	Copper3D (Santiago, Chile)	PLA with Nano-Copper additive	16708001
F	PrimaSelect PLA AntiBac	PrimaCreator (Malmö, Sweden)	Based on PLA, manufacturer doesn’t specify additives	FB0195
G	Smartfil	SMART MATERIALS 3D (Alcalá la Real (Jaén), Spain)	PLA with silver nanoparticles	129417002085
H	Tarfuse^®^ PLA AM	Grupa Azoty S.A. (Tarnów, Polska)	Based on PLA, manufacturer doesn’t specify additives	N/S *
I	Antibacterial PLA	XYZ Printing Inc. (New Taipei City, Taiwan)	PLA with silver additives	RFPLK-FPE-B6W-TH-92K-0364

* manufacturer does not specify identification number.

**Table 2 polymers-13-04337-t002:** Non-variable manufacturing process parameters and material specific parameters.

Process Parameters				
Layer Thickness (mm)	Infill (%)	Cooling Fan Speed (%)	Perimeter Speed (mm/s	Infill Speed (mm/s)
0.2	100	100	45	80
Material Specific Process Parameters				
Material Name	Printhead Temperature (°C)		Bed Temperature (°C)	
A	210		50	
B	210		55	
C	200		60	
D	210		60	
E	200		60	
F	210		50	
G	220		60	
H	220		60	
I	205		50	

The process parameters were selected based on typical values for PLA material and the recommendations of the manufacturers of individual materials.

**Table 3 polymers-13-04337-t003:** Results of 2D digital radiography of filaments and samples manufactured additively.

Material Name	RTG of Filament	RTG of FFF Sample
A	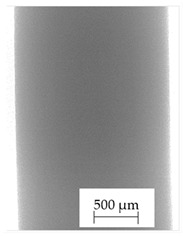	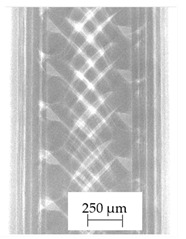
B	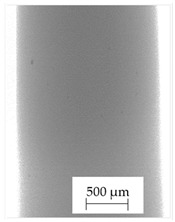	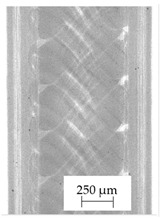
C	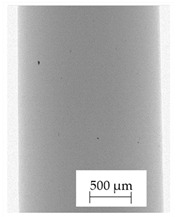	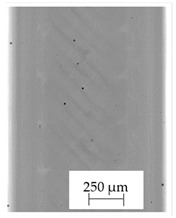
D	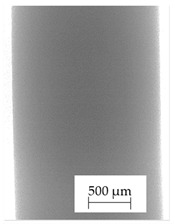	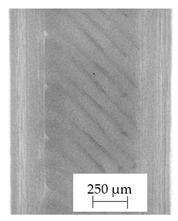
E	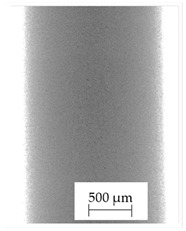	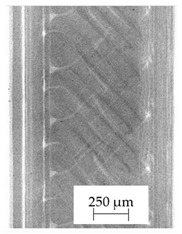
F	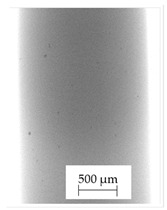	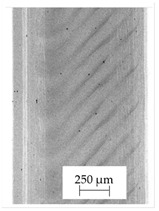
G	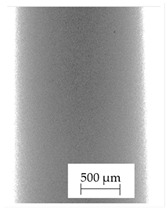	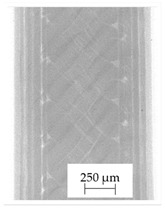
H	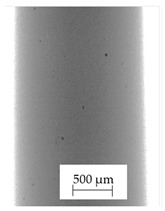	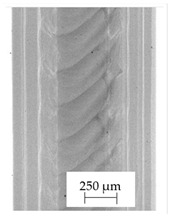
I	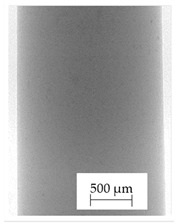	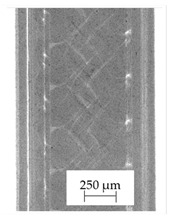

**Table 4 polymers-13-04337-t004:** Results of SEM–EDS analysis—the presence of specific elements in analyzed micro-areas of filaments.

Material Name	Elements											
	C	O	Al	Si	Ag	Ca	Cu	P	Mg	Na	Ti	Cl
A	66.55 ± 16.25	19.3 ± 9.8	0.45± 0.05	6.35 ± 5.55	-	6.70 ± 0.01	-	-	0.50 ± 0.01	1.10 ± 0.01	6.50± 0.01	-
B	88.37 ± 5.76	5.40 ± 1.49	0.30 ± 0.20	0.70 ± 0.16	-	-	-	-	0.10 ± 0.01	0.15± 0.05	1.10 ± 0.20	13.20 ± 0.10
C	90.50 ± 0.92	7.73 ± 0.75	-	0.60 ± 0.01	-	1.50 ± 0.01	0.10 ± 0.01	1.10 ± 0.01	0.40 ± 0.01	-	-	-
D	91.50 ± 1.28	7.37 ± 1.22	0.30 ± 0.08	0.67 ± 0.24	-	-	-	-	0.25 ± 0.05	0.20 ± 0.01	-	-
E	90.03 ± 1.27	7.97 ± 0.73	0.90 ± 0.22	0.83 ± 0.17	-	-	0.50 ± 0.10	-	-	-	-	-
F	91.43 ± 1.47	6.73 ± 0.90	0.43 ± 0.05	0.40 ± 0.08	-	-	-	-	0.30 ± 0.10	0.40 ± 0.01	2.10 ± 0.01	-
G	76.17 ± 9.38	9.10 ± 0.21	1.16 ± 0.24	4.36 ± 2.64	0.03 ± 0.05	9.13 ± 10.9	-	-	-	-	-	-
H	88.17 ± 6.41	6.70 ± 0.22	2.60 ± 3.18	2.53 ± 2.95	-	-	0.10 ± 0.01	-	-	-	-	-
I	86.64 ± 4.52	10.60 ± 3.29	0.40 ± 0.22	0.80 ± 0.36	-	-	-	-	1.20 ± 0.54	1.20 ± 0.01	-	-
J	3.52 ± 0.68	29.17 ± 3.50	3.74 ± 0.23	2.56 ± 0.11	-	-	57.65 ± 1.58	-	-	-	-	4.33 ± 0.23

**Table 5 polymers-13-04337-t005:** A study based on the manufacturers’ report on the antimicrobial activity of filaments. Letters A–I: specific type of material analyzed.

Material Name	Producer Report According to Antimicrobial Activity
A	N/A
B	N/A
C	Method based on ISO 22196. Effectiveness on *E.* *coli* ATCC 8739 after 8 h—99.97181% Effectiveness on *E.* *coli* ATCC 8739 after 24 h—99.98909%
D	N/A
E	Effectiveness on *S. aureus* MRSA after 8 h >98%; after 24 h > 99.99% Effectiveness on *E.* *coli* DH5 after 8 h > 98% after 24 h > 99.99%.
F	Method based on ISO 22196. Effectiveness on *S.* *aureus* after 24 h 99.59% Effectiveness on *E.* *coli* DH5 after 24 h 88.43%
G	Method based on JIS Z 2801 (ISO 22196). Effectiveness on *S.* *aureus* CECT 240, ATCC 6538 P after 24 h—99.99% Effectiveness on *E.* *coli* CECT 516, ATCC 8739 after 24 h—99.99%
H	The antibacterial additives used in the H filament are approved for marketing in the European Union—they comply with the European Regulation on biocidal products (BPR, Regulation (EU) 528/2012 and with the requirements of the American Environmental Protection Agency—Antimicrobial Division of the Environmental Protection Agency (EPA).The antibacterial additives used are included in the list of chemical compounds approved by OEKO-TEX.
I	N/A

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
