# Peer review of "Material Extrusion-Based Additive Manufacturing of Poly(Lactic Acid) Antibacterial Filaments—A Case Study of Antimicrobial Properties"

_polymers, 2021, doi:10.3390/polym13244337_

Round 1

Reviewer 1 Report

The manuscript is very well presented but needs minor modification.

  1. Extend the introduction section and add more literature. Highlight the novelty part of the manuscript in the last paragraph of the manuscript. 
  2. What is the rationale behind the selection of input process parameters? Discuss the same.
  3. Index the SEM images with appropriate features.
  4. Interpret the results with literature review and past work.
  5. Re-write the conclusion and highlight major findigs.

Author Response

Dear Reviewer,
The responses to the review are attached as a file.
Thank you for the positive reception of our manuscript.

Reviewer 2 Report

This research provides interesting results for the antimicrobial polymers. However, more discussion and explanation should be added to strengthen the novelty of the research.

Abstract should be revised to strengthen the novelty or importance of the finding/results. Half of the abstract is the introduction. There should be more statement about the results.

L 27 should specifically mention which system? And add short discussion/explanation of the key results.

Introduction

At the beginning, the introduction state about the COVID pandemic which make the reader felt confused about the antimicrobial test. As Covid is virus but the antimicrobial testing was in bacteria. There should be statement about the importance of this antibacterial test, e.g., reason for testing these microorganisms. There are also some discussions of the applications of these PLA/antimicrobial polymers. Are these microorganisms available on those mentioned applications?

There can be statements on the concern of safety due to the presence of these microorganisms.

L61 Why it is important to verify? The objective/aim of the research should be specifically mentioned in this paragraph.

Results and discussion

L146-147 The sentence is difficult to understand, please revise.

L150 Please add discussion why the FDM gave disconnection structure.

Table 3 The scale bars differed between the columns. Does this mean different method provided different material thickness/size?

SEM section. There should be discussion/explanation/importance of SEM experiments. The section has too few discussions.

L199 Does this mean the antimicrobial capacity was lost (at least partial) after the FDM processing? If so, please discuss the reason.

L212-219 Please discuss how the PLA filaments can be applied with this discussion.

Conclusions should answer or support the question/objective/goal of the research. Please consider to add conclusions that was supported by the present data.

There is mention about the use of this antimicrobial polymers in the hospital. How the results of these microorganisms can be applied, e.g. are they typically present on surface of the hospital stuffs?

Author Response

(The authors gave the same response as above.)

Reviewer 3 Report

The submitted paper presents the results of the research entitled "Material extrusion-based additive manufacturing of polylactic acid antibacterial filaments – case study of antimicrobial properties". The subject of the work tries to refer to the quite popular trend of research on COVID pandemic, but in my opinion, the presented results are not suitable for publication. Some comments are highlighted below.

  1. The research was focused "only" on antibacterial properties of the investigated PLA filaments, while in my opinion other properties might be also important (mechanical, thermomechanical....)
  2. In my opinion, the selection of materials for the tests is very wrong, the authors prepared samples from a fairly wide range of filamants available on the market, without analyzing the composition, content of the filler or even its type.
  3. In this case, the analysis of antibacterial properties does not bring any new information, it is only an indication whether a given filament from a given batch is effective. What if the producer decided to change the composition of the materials.
  4. A large proportion of the results do not bring any relevant information. for example, the analysis of images of the sample structure allows only to observe the presence of voids in the sample, without the density analysis such information is of little importance. The same conclusion refer to the results of SEM-BSE analysis, where the authors claim that they identify filler particles. In my opinion, without identification by means of the SEM-EDS analysis, it is difficult to say whether the observed particles are not contaminants.

Some suggestions for a possible correction of the article:

I suggest you prepare your own materials based on the known composition, including the type of used antibacterial additives. Depending on the concept, the authors can then estimate the antimicrobial effectiveness depending on the content of a certain additive or different types of additive.

Author Response

Dear Reviewer,
The responses to the review are attached as a file.

Round 2

Reviewer 2 Report

The manuscript has been revised and improved.

Author Response

We would like to thank you again for the positive reception of our manuscript. Regarding the version sent to the reviewer, and referring to the latest minor revisions, changes were made to the manuscript as required by the Academic Editor. 

Reviewer 3 Report

Despite significant additions to the text, the work still does not meet the requirements of the scientific article. The introduced supplements do not bring any new important information to the work, what is interesting, despite the significant amount of the new text, the literature has not been supplemented. I can support the previous assessment and reject the submitted paper.

Author Response

Thanks for your comments. In the first round of the reviews we have expanded the scope of the manuscript for additional research, improved the quality of the introduction and discussion part, and all the new information added was proven with essential references. Additionally, according to the Academic Editor suggestions, we have introduced changes into the introduction. It resulted also in addition of 6 more positions, which gives a total increase of 19 references compared with the first original draft. 
